# On Optimal Generalizability in Parametric Learning

**Ahmad Beirami**[*]
beirami@seas.harvard.edu

**Meisam Razaviyayn**[†]
razaviya@usc.edu

**Shahin Shahrampour**[*]
shahin@seas.harvard.edu

**Vahid Tarokh**[*]
vahid@seas.harvard.edu

## Abstract

We consider the parametric learning problem, where the objective of the learner is determined by a parametric loss function. Employing empirical risk minimization with possibly regularization, the inferred parameter vector will be biased toward the training samples. Such bias is measured by the cross validation procedure in practice where the data set is partitioned into a training set used for training and a validation set, which is not used in training and is left to measure the out-of-sample performance. A classical cross validation strategy is the leave-one-out cross validation (LOOCV) where one sample is left out for validation and training is done on the rest of the samples that are presented to the learner, and this process is repeated on all of the samples. LOOCV is rarely used in practice due to the high computational complexity. In this paper, we first develop a computationally efficient approximate LOOCV (ALOOCV) and provide theoretical guarantees for its performance. Then we use ALOOCV to provide an optimization algorithm for finding the regularizer in the empirical risk minimization framework. In our numerical experiments, we illustrate the accuracy and efficiency of ALOOCV as well as our proposed framework for the optimization of the regularizer.

## 1 Introduction

We consider the parametric supervised/unsupervised learning problem, where the objective of the learner is to build a predictor based on a set of historical data. Let $z^n = \{z_i\}_{i=1}^n$, where $z_i \in \mathcal{Z}$ denotes the data samples at the learner's disposal that are assumed to be drawn i.i.d. from an unknown density function $p(\cdot)$, and $\mathcal{Z}$ is compact.

We assume that the learner expresses the objective in terms of minimizing a parametric loss function $\ell(z; \boldsymbol{\theta})$, which is a function of the parameter vector $\boldsymbol{\theta}$. The learner solves for the unknown parameter vector $\boldsymbol{\theta} \in \Theta \subseteq \mathbb{R}^k$, where $k$ denotes the number of parameters in the model class, and $\Theta$ is a convex, compact set.

Let

$$\mathcal{L}(\boldsymbol{\theta}) \triangleq E\{\ell(z; \boldsymbol{\theta})\} \tag{1}$$

be the risk associated with the parameter vector $\boldsymbol{\theta}$, where the expectation is with respect to the density $p(\cdot)$ that is unknown to the learner. Ideally, the goal of the learner is to choose the parameter vector $\boldsymbol{\theta}^*$ such that $\boldsymbol{\theta}^* \in \arg\min_{\boldsymbol{\theta} \in \Theta} \mathcal{L}(\boldsymbol{\theta}) = \arg\min_{\boldsymbol{\theta} \in \Theta} E\{\ell(z; \boldsymbol{\theta})\}$. Since the density function $p(\cdot)$ is unknown, the learner cannot compute $\boldsymbol{\theta}^*$ and hence cannot achieve the ideal performance of $\mathcal{L}(\boldsymbol{\theta}^*) = \min_{\boldsymbol{\theta} \in \Theta} \mathcal{L}(\boldsymbol{\theta})$ associated with the model class $\Theta$. Instead, one can consider the minimiza-

---

[*]School of Engineering and Applied Sciences, Harvard University, Cambridge, MA 02138, USA.

[†]Department of Industrial and Systems Engineering, University of Southern California, Los Angeles, CA 90089, USA.

tion of the empirical version of the problem through the empirical risk minimization framework:

$$\widehat{\boldsymbol{\theta}}(z^n) \in \arg\min_{\boldsymbol{\theta} \in \Theta} \quad \sum_{i \in [n]} \ell(z_i; \boldsymbol{\theta}) + r(\boldsymbol{\theta}),$$

where $[n] \triangleq \{1, 2, \ldots, n\}$ and $r(\boldsymbol{\theta})$ is some regularization function. While the learner can evaluate her performance on the training data samples (also called the in-sample empirical risk, i.e., $\frac{1}{n} \sum_{i=1}^{n} \ell(z_i; \widehat{\boldsymbol{\theta}}(z^n))$), it is imperative to assess the average performance of the learner on fresh test samples, i.e., $\mathcal{L}(\widehat{\boldsymbol{\theta}}(z^n))$, which is referred to as the out-of-sample risk. A simple and universal approach to measuring the out-of-sample risk is cross validation [1]. Leave-one-out cross validation (LOOCV), which is a popular exhaustive cross validation strategy, uses $(n-1)$ of the samples for training while one sample is left out for testing. This procedure is repeated on the $n$ samples in a round-robin fashion, and the learner ends up with $n$ estimates for the out-of-sample loss corresponding to each sample. These estimates together form a cross validation vector which can be used for the estimation of the out-of-sample performance, model selection, and tuning the model hyperparameters. While LOOCV provides a reliable estimate of the out-of-sample loss, it brings about an additional factor of $n$ in terms of computational cost, which makes it practically impossible because of the high computational cost of training when the number of samples is large.

**Contribution:** Our first contribution is to provide an approximation for the cross validation vector, called ALOOCV, with much lower computational cost. We compare its performance with LOOCV in problems of reasonable size where LOOCV is tractable. We also test it on problems of large size where LOOCV is practically impossible to implement. We describe how to handle quasi-smooth loss/regularizer functions. We also show that ALOOCV is asymptotically equivalent to Takeuchi information criterion (TIC) under certain regularity conditions.

Our second contribution is to use ALOOCV to develop a gradient descent algorithm for jointly optimizing the regularization hyperparameters as well as the unknown parameter vector $\boldsymbol{\theta}$. We show that multiple hyperparameters could be tuned using the developed algorithm. We emphasize that the second contribution would not have been possible without the developed estimator as obtaining the gradient of the LOOCV with respect to tuning parameters is computationally expensive. Our experiments show that the developed method handles quasi-smooth regularized loss functions as well as number of tuning parameters that is on the order of the training samples.

Finally, it is worth mentioning that although the leave-one-out cross validation scenario is considered in our analyses, the results and the algorithms can be extended to the leave-$q$-out cross validation and bootstrap techniques.

**Related work:** A main application of cross validation (see [1] for a recent survey) is in model selection [2–4]. On the theoretical side, the proposed approximation on LOOCV is asymptotically equivalent to Takeuchi information criterion (TIC) [4–7], under certain regularity conditions (see [8] for a proof of asymptotic equivalence of AIC and LOOCV in autoregressive models). This is also related to Barron's predicted square error (PSE) [9] and Moody's effective number of parameters for nonlinear systems [10]. Despite these asymptotic equivalences our main focus is on the non-asymptotic performance of ALOOCV.

ALOOCV simplifies to the closed form derivation of the LOOCV for linear regression, called PRESS (see [11, 12]). Hence, this work can be viewed as an approximate extension of this closed form derivation for an arbitrary smooth regularized loss function. This work is also related to the concept of influence functions [13], which has recently received renewed interest [14]. In contrast to methods based on influence functions that require large number of samples due to their asymptotic nature, we empirically show that the developed ALOOCV works well even when the number of samples and features are small and comparable to each other. In particular, ALOOCV is capable of predicting overfitting and hence can be used for model selection and choosing the regularization hyperparameter. Finally, we expect that the idea of ALOOCV can be extended to derive computationally efficient approximate bootstrap estimators [15].

Our second contribution is a gradient descent optimization algorithm for tuning the regularization hyperparameters in parametric learning problems. A similar approach has been taken for tuning the single parameter in ridge regression where cross validation can be obtained in closed form [16]. Most of the existing methods, on the other hand, ignore the response and carry out the optimization solely based on the features, e.g., Stein unbiased estimator of the risk for multiple parameter selection [17, 18].

Bayesian optimization has been used for tuning the hyperparameters in the model [19–23], which postulates a prior on the parameters and optimizes for the best parameter. Bayesian optimization methods are generally derivative free leading to slow convergence rate. In contrast, the proposed method is based on a gradient descent method. Other popular approaches to the tuning of the optimization parameters include grid search and random search [24–26]. These methods, by nature, also suffer from slow convergence. Finally, model selection has been considered as a bi-level optimization [27,28] where the training process is modeled as a second level optimization problem within the original problem. These formulations, similar to many other bi-level optimization problems, often lead to computationally intensive algorithms that are not scalable.

We remark that ALOOCV can also be used within Bayesian optimization, random search, and grid search methods. Further, resource allocation can be used for improving the optimization performance in all of such methods.

## 2  Problem Setup

To facilitate the presentation of the ideas, let us define the following concepts. Throughout, we assume that all the vectors are in column format.

**Definition 1 (regularization vector/regularized loss function)** *We suppose that the learner is concerned with $M$ regularization functions $r_1(\boldsymbol{\theta}), \ldots, r_M(\boldsymbol{\theta})$ in addition to the main loss function $\ell(z; \boldsymbol{\theta})$. We define the regularization vector $\boldsymbol{r}(\boldsymbol{\theta})$ as*

$$\boldsymbol{r}(\boldsymbol{\theta}) \triangleq (r_1(\boldsymbol{\theta}), \ldots, r_M(\boldsymbol{\theta}))^\top.$$

*Further, let $\boldsymbol{\lambda} = (\lambda_1, \ldots, \lambda_M)^\top$ be the vector of regularization parameters. We call $w_n(z; \boldsymbol{\theta}, \boldsymbol{\lambda})$ the regularized loss function given by*

$$w_n(z; \boldsymbol{\theta}, \boldsymbol{\lambda}) \triangleq \ell(z; \boldsymbol{\theta}) + \frac{1}{n} \boldsymbol{\lambda}^\top \boldsymbol{r}(\boldsymbol{\theta}) = \ell(z; \boldsymbol{\theta}) + \frac{1}{n} \sum_{m \in [M]} \lambda_m r_m(\boldsymbol{\theta}).$$

The above definition encompasses many popular learning problems. For example, elastic net regression [31] can be cast in this framework by setting $r_1(\boldsymbol{\theta}) = \|\boldsymbol{\theta}\|_1$ and $r_2(\boldsymbol{\theta}) = \frac{1}{2}\|\boldsymbol{\theta}\|_2^2$.

**Definition 2 (empirical risk/regularized empirical risk)** *Let the empirical risk be defined as $\widehat{\mathcal{L}}_{z^n}(\boldsymbol{\theta}) = \frac{1}{n} \sum_{i=1}^n \ell(z_i; \boldsymbol{\theta})$. Similarly, let the regularized empirical risk be defined as $\widehat{\mathcal{W}}_{z^n}(\boldsymbol{\theta}, \boldsymbol{\lambda}) = \frac{1}{n} \sum_{i=1}^n \{w_n(z_i; \boldsymbol{\theta}, \boldsymbol{\lambda})\}$.*

**Definition 3 (regularized empirical risk minimization)** *We suppose that the learner solves the empirical risk minimization problem by selecting $\widehat{\boldsymbol{\theta}}_{\boldsymbol{\lambda}}(z^n)$ as follows:*

$$\widehat{\boldsymbol{\theta}}_{\boldsymbol{\lambda}}(z^n) \in \arg\min_{\boldsymbol{\theta} \in \Theta} \left\{ \widehat{\mathcal{W}}_{z^n}(\boldsymbol{\theta}, \boldsymbol{\lambda}) \right\} = \arg\min_{\boldsymbol{\theta} \in \Theta} \left\{ \sum_{i \in [n]} \ell(z_i; \boldsymbol{\theta}) + \boldsymbol{\lambda}^\top \boldsymbol{r}(\boldsymbol{\theta}) \right\}. \qquad (2)$$

Once the learner solves for $\widehat{\boldsymbol{\theta}}_{\boldsymbol{\lambda}}(z^n)$, the empirical risk corresponding to $\widehat{\boldsymbol{\theta}}_{\boldsymbol{\lambda}}(z^n)$ can be readily computed by $\widehat{\mathcal{L}}_{z^n}(\widehat{\boldsymbol{\theta}}_{\boldsymbol{\lambda}}(z^n)) = \frac{1}{n} \sum_{i \in [n]} \ell(z_i; \widehat{\boldsymbol{\theta}}_{\boldsymbol{\lambda}}(z^n))$. While the learner can evaluate her performance on the observed data samples (also called the in-sample empirical risk, i.e., $\widehat{\mathcal{L}}_{z^n}(\widehat{\boldsymbol{\theta}}_{\boldsymbol{\lambda}}(z^n))$), it is imperative to assess the performance of the learner on unobserved fresh samples, i.e., $\mathcal{L}(\widehat{\boldsymbol{\theta}}_{\boldsymbol{\lambda}}(z^n))$ (see (1)), which is referred to as the out-of-sample risk. To measure the out-of-sample risk, it is a common practice to perform cross validation as it works outstandingly well in many practical situations and is conceptually universal and simple to implement.

Leave-one-out cross validation (LOOCV) uses all of the samples but one for training, which is left out for testing, leading to an $n$-dimensional cross validation vector of out-of-sample estimates. Let us formalize this notion. Let $z^{n \backslash i} \triangleq (z_1, \ldots, z_{i-1}, z_{i+1}, \ldots, z_n)$ denote the set of the training examples excluding $z_i$.

**Definition 4 (LOOCV empirical risk minimization/cross validation vector)** *Let* $\widehat{\boldsymbol{\theta}}_{\boldsymbol{\lambda}}(z^{n\setminus i})$ *be the estimated parameter over the training set* $z^{n\setminus i}$, *i.e.,*

$$\widehat{\boldsymbol{\theta}}_{\boldsymbol{\lambda}}(z^{n\setminus i}) \in \arg\min_{\boldsymbol{\theta}\in\mathbb{R}^k} \left\{\widehat{\mathcal{W}}_{z^{n\setminus i}}(\boldsymbol{\theta},\boldsymbol{\lambda})\right\} = \arg\min_{\boldsymbol{\theta}\in\mathbb{R}^k} \left\{\sum_{j\in[n]\setminus i} \ell(z_j;\boldsymbol{\theta}) + \boldsymbol{\lambda}^\top \boldsymbol{r}(\boldsymbol{\theta})\right\}. \qquad (3)$$

*The cross validation vector is given by* $\{CV_{\boldsymbol{\lambda},i}(z^n)\}_{i\in[n]}$ *where* $CV_{\boldsymbol{\lambda},i}(z^n) \triangleq \ell(z_i;\widehat{\boldsymbol{\theta}}_{\boldsymbol{\lambda}}(z^{n\setminus i}))$, *and the cross validation out-of-sample estimate is given by* $\overline{CV}_{\boldsymbol{\lambda}}(z^n) \triangleq \frac{1}{n}\sum_{i=1}^n CV_{\boldsymbol{\lambda},i}(z^n)$.

The empirical mean and the empirical variance of the $n$-dimensional cross validation vector are used by practitioners as surrogates on assessing the out-of-sample performance of a learning method.

The computational cost of solving the problem in (3) is $n$ times that of the original problem in (2). Hence, while LOOCV provides a simple yet powerful tool to estimate the out-of-sample performance, the additional factor of $n$ in terms of the computational cost makes LOOCV impractical in large-scale problems. One common solution to this problem is to perform validation on fewer number of samples, where the downside is that the learner would obtain a much more noisy and sometimes completely unstable estimate of the out-of-sample performance compared to the case where the entire LOOCV vector is at the learner's disposal. On the other hand, ALOOCV described next will provide the benefits of LOOCV with negligible additional computational cost.

We emphasize that the presented problem formulation is general and includes a variety of parametric machine learning tasks, where the learner empirically solves an optimization problem to minimize some loss function.

## 3 Approximate Leave-One-Out Cross Validation (ALOOCV)

We assume that the regularized loss function is three times differentiable with continuous derivatives (see Assumption 1). This includes many learning problems, such as the $L_2$ regularized logistic loss function. We later comment on how to handle the $\ell_1$ regularizer function in LASSO. To proceed, we need one more definition.

**Definition 5 (Hessian/empirical Hessian)** *Let* $\mathcal{H}(\boldsymbol{\theta})$ *denote the Hessian of the risk function defined as* $\mathcal{H}(\boldsymbol{\theta}) \triangleq \nabla^2_{\boldsymbol{\theta}}\mathcal{L}(\boldsymbol{\theta})$. *Further, let* $\widehat{\mathcal{H}}_{z^n}(\boldsymbol{\theta},\boldsymbol{\lambda})$ *denote the empirical Hessian of the regularized loss function, defined as* $\widehat{\mathcal{H}}_{z^n}(\boldsymbol{\theta},\boldsymbol{\lambda}) \triangleq \widehat{E}_{z^n}\left\{\nabla^2_{\boldsymbol{\theta}} w_n(z;\boldsymbol{\theta},\boldsymbol{\lambda})\right\} = \frac{1}{n}\sum_{i=1}^n \nabla^2_{\boldsymbol{\theta}} w_n(z_i;\boldsymbol{\theta},\boldsymbol{\lambda})$. *Similarly, we define* $\widehat{\mathcal{H}}_{z^n}(\boldsymbol{\theta},\boldsymbol{\lambda}) \triangleq \widehat{E}_{z^{n\setminus i}}\left\{\nabla^2_{\boldsymbol{\theta}} w_n(z;\boldsymbol{\theta},\boldsymbol{\lambda})\right\} = \frac{1}{n-1}\sum_{i\in[n]\setminus i} \nabla^2_{\boldsymbol{\theta}} w_n(z_i;\boldsymbol{\theta},\boldsymbol{\lambda})$.

Next we present the set of assumptions we need to prove the main result of the paper.

**Assumption 1** *We assume that*

*(a) There exists* $\boldsymbol{\theta}^* \in \Theta^\circ$,[3] *such that* $\|\widehat{\boldsymbol{\theta}}_{\boldsymbol{\lambda}}(z^n) - \boldsymbol{\theta}^*\|_\infty = o_p(1)$.[4]

*(b)* $w_n(z;\boldsymbol{\theta})$ *is of class* $C^3$ *as a function of* $\boldsymbol{\theta}$ *for all* $z \in \mathcal{Z}$.

*(c)* $\mathcal{H}(\boldsymbol{\theta}^*) \succ 0$ *is positive definite.*

**Theorem 1** *Under Assumption 1, let*

$$\widetilde{\boldsymbol{\theta}}_{\boldsymbol{\lambda}}^{(i)}(z^n) \triangleq \widehat{\boldsymbol{\theta}}_{\boldsymbol{\lambda}}(z^n) + \frac{1}{n-1}\left(\widehat{\mathcal{H}}_{z^{n\setminus i}}\left(\widehat{\boldsymbol{\theta}}_{\boldsymbol{\lambda}}(z^n),\boldsymbol{\lambda}\right)\right)^{-1}\nabla_{\boldsymbol{\theta}}\ell(z_i;\widehat{\boldsymbol{\theta}}_{\boldsymbol{\lambda}}(z^n)), \qquad (4)$$

*assuming the inverse exists. Then,*

$$\widehat{\boldsymbol{\theta}}_{\boldsymbol{\lambda}}(z^{n\setminus i}) - \widetilde{\boldsymbol{\theta}}_{\boldsymbol{\lambda}}^{(i)}(z^n) = \frac{1}{n-1}\left(\widehat{\mathcal{H}}_{z^{n\setminus i}}\left(\widehat{\boldsymbol{\theta}}_{\boldsymbol{\lambda}}(z^n),\boldsymbol{\lambda}\right)\right)^{-1}\varepsilon_{\boldsymbol{\lambda},n}^{(i)}, \qquad (5)$$

*with high probability where*

$$\varepsilon_{\boldsymbol{\lambda},n}^{(i)} = \varepsilon_{\boldsymbol{\lambda},n}^{(i),1} - \varepsilon_{\boldsymbol{\lambda},n}^{(i),2}, \tag{6}$$

*and $\varepsilon_{\boldsymbol{\lambda},n}^{(i),1}$ is defined as*

$$\varepsilon_{\boldsymbol{\lambda},n}^{(i),1} \triangleq \frac{1}{2} \sum_{j \in [n] \setminus i} \sum_{\kappa \in [k]} (\widehat{\boldsymbol{\theta}}_{\boldsymbol{\lambda}}(z^n) - \widehat{\boldsymbol{\theta}}_{\boldsymbol{\lambda}}(z^{n \setminus i}))^\top \left( \frac{\partial}{\partial \theta_\kappa} \nabla_{\boldsymbol{\theta}}^2 w_{n-1}(z_j; \boldsymbol{\zeta}_{\boldsymbol{\lambda},\kappa}^{i,j,1}(z^n), \boldsymbol{\lambda}) \right) (\widehat{\boldsymbol{\theta}}_{\boldsymbol{\lambda}}(z^n) - \widehat{\boldsymbol{\theta}}_{\boldsymbol{\lambda}}(z^{n \setminus i})) \widehat{e}_\kappa,$$

$$\tag{7}$$

*where $\widehat{e}_\kappa$ is $\kappa$-th standard unit vector, and such that for all $\kappa \in [k]$, $\boldsymbol{\zeta}_{\boldsymbol{\lambda},\kappa}^{i,j,1}(z^n) = \alpha_\kappa^{i,j,1} \widehat{\boldsymbol{\theta}}_{\boldsymbol{\lambda}}(z^n) + (1 - \alpha_\kappa^{i,j,1}) \widehat{\boldsymbol{\theta}}_{\boldsymbol{\lambda}}(z^{n \setminus i})$ for some $0 \leq \alpha_\kappa^{i,j,1} \leq 1$. Further, $\varepsilon_{\boldsymbol{\lambda},n}^{(i),2}$ is defined as*

$$\varepsilon_{\boldsymbol{\lambda},n}^{(i),2} \triangleq \sum_{j \in [n] \setminus i} \sum_{\kappa,\nu \in [k]} \widehat{e}_\nu^\top (\widehat{\boldsymbol{\theta}}_{\boldsymbol{\lambda}}(z^n) - \widehat{\boldsymbol{\theta}}_{\boldsymbol{\lambda}}(z^{n \setminus i})) \left( \frac{\partial^2}{\partial \theta_\kappa \partial \theta_\nu} \nabla_{\boldsymbol{\theta}}^\top w_{n-1}(z_j; \boldsymbol{\zeta}_{\boldsymbol{\lambda},\kappa,\nu}^{i,j,2}(z^n), \boldsymbol{\lambda}) \right) (\widehat{\boldsymbol{\theta}}_{\boldsymbol{\lambda}}(z^n) - \widehat{\boldsymbol{\theta}}_{\boldsymbol{\lambda}}(z^{n \setminus i})) \widehat{e}_\kappa,$$

$$\tag{8}$$

*such that for $\kappa, \nu \in [k]$, $\boldsymbol{\zeta}_{\boldsymbol{\lambda},\kappa,\nu}^{(i),2}(z^n) = \alpha_{\kappa,\nu}^{i,j,2} \widehat{\boldsymbol{\theta}}_{\boldsymbol{\lambda}}(z^n) + (1 - \alpha_{\kappa,\nu}^{i,j,2}) \widehat{\boldsymbol{\theta}}_{\boldsymbol{\lambda}}(z^{n \setminus i})$ for some $0 \leq \alpha_{\kappa,\nu}^{i,j,2} \leq 1$. Further, we have[5]*

$$\|\widehat{\boldsymbol{\theta}}_{\boldsymbol{\lambda}}(z^n) - \widehat{\boldsymbol{\theta}}_{\boldsymbol{\lambda}}(z^{n \setminus i}))\|_\infty = O_p \left( \frac{1}{n} \right), \tag{9}$$

$$\|\widehat{\boldsymbol{\theta}}_{\boldsymbol{\lambda}}(z^{n \setminus i}) - \widetilde{\boldsymbol{\theta}}_{\boldsymbol{\lambda}}^{(i)}(z^n)\|_\infty = O_p \left( \frac{1}{n^2} \right). \tag{10}$$

See the appendix for the proof. Inspired by Theorem 1, we provide an approximation on the cross validation vector.

**Definition 6 (approximate cross validation vector)** *Let $ACV_{\boldsymbol{\lambda},i}(z^n) = \ell \left( z_i; \widetilde{\boldsymbol{\theta}}_{\boldsymbol{\lambda}}^{(i)}(z^n) \right)$. We call $\{ACV_{\boldsymbol{\lambda},i}(z^n)\}_{i \in [n]}$ the approximate cross validation vector. We further call*

$$\overline{ACV}_{\boldsymbol{\lambda}}(z^n) \triangleq \frac{1}{n} \sum_{i=1}^n ACV_{\boldsymbol{\lambda},i}(z^n) \tag{11}$$

*the approximate cross validation estimator of the out-of-sample loss.*

We remark that the definition can be extended to leave-$q$-out and $q$-fold cross validation by replacing the index $i$ to an index set $S$ with $|S| = q$, comprised of the $q$ left-out samples in (4).

The cost of the computation of $\{\widetilde{\boldsymbol{\theta}}_{\boldsymbol{\lambda}}^{(i)}(z^n)\}_{i \in [n]}$ is upper bounded by $O(np + C(n,p))$ where $C(n,p)$ is the computational cost of solving for $\widehat{\boldsymbol{\theta}}_{\boldsymbol{\lambda}}(z^n)$ in (2); see [14]. Note that the empirical risk minimization problem posed in (2) requires time at least $\Omega(np)$. Hence, the overall cost of computation of $\{\widetilde{\boldsymbol{\theta}}_{\boldsymbol{\lambda}}^{(i)}(z^n)\}_{i \in [n]}$ is dominated by solving (2). On the other hand, the cost of computing the true cross validation performance by naively solving $n$ optimization problems $\{\widehat{\boldsymbol{\theta}}_{\boldsymbol{\lambda}}(z^{n \setminus i})\}_{i \in [n]}$ posed in (3) would be $O(nC(n,p))$ which would necessarily be $\Omega(n^2 p)$ making it impractical for large-scale problems.

**Corollary 2** *The approximate cross validation vector is exact for kernel ridge regression. That is, given that the regularized loss function is quadratic in $\boldsymbol{\theta}$, we have $\widetilde{\boldsymbol{\theta}}_{\boldsymbol{\lambda}}^{(i)}(z^n) = \widehat{\boldsymbol{\theta}}_{\boldsymbol{\lambda}}(z^{n \setminus i})$ for all $i \in [n]$.*

**Proof** We notice that the error term $\varepsilon_{\boldsymbol{\lambda},n}^{(i)}$ in (6) only depends on the third derivative of the loss function in a neighborhood of $\widehat{\boldsymbol{\theta}}_{\boldsymbol{\lambda}}(z^n)$. Hence, provided that the regularized loss function is quadratic in $\boldsymbol{\theta}$, $\varepsilon_{\boldsymbol{\lambda},n}^{(i)} = 0$ for all $i \in [n]$. ∎

The fact that the cross validation vector could be obtained for kernel ridge regression in closed form without actually performing cross validation is not new, and the method is known as PRESS [11]. In a sense, the presented approximation could be thought of as an extension of this idea to more general loss and regularizer functions while losing the exactness property. We remark that the idea of ALOOCV is also related to that of the influence functions. In particular, influence functions have been used in [14] to derive an approximation on LOOCV for neural networks with large sample sizes. However, we notice that methods based on influence functions usually underestimate overfitting making them impractical for model selection. In contrast, we empirically demonstrate the effectiveness of ALOOCV in capturing overfitting and model selection.

In the case of $\ell_1$ regularizer we assume that the support set of $\widehat{\boldsymbol{\theta}}_{\boldsymbol{\lambda}}(z^n)$ and $\widehat{\boldsymbol{\theta}}_{\boldsymbol{\lambda}}(z^{n\backslash i})$ are the same. Although this would be true for large enough $n$ under Assumption 1, it is not necessarily true for a given sample $z^n$ when sample $i$ is left out. Provided that the support set of $\widehat{\boldsymbol{\theta}}_{\boldsymbol{\lambda}}(z^{n\backslash i})$ is known we use the developed machinery in Theorem 1 on the subset of parameters that are non-zero. Further, we ignore the $\ell_1$ regularizer term in the regularized loss function as it does not contribute to the Hessian matrix locally, and we assume that the regularized loss function is otherwise smooth in the sense of Assumption 1. In this case, the cost of calculating ALOOCV would scale with $O(np_a \log(1/\epsilon))$ where $p_a$ denotes the number of non-zero coordinates in the solution $\widehat{\boldsymbol{\theta}}_{\boldsymbol{\lambda}}(z^n)$.

We remark that although the nature of guarantees in Theorem 1 are asymptotic, we have experimentally observed that the estimator works really well even for $n$ and $p$ as small as $50$ in elastic net regression, logistic regression, and ridge regression. Next, we also provide an asymptotic characterization of the approximate cross validation.

**Lemma 3** *Under Assumption 1, we have*

$$\overline{ACV}_{\boldsymbol{\lambda}}(z^n) = \widehat{\mathcal{L}}_{z^n}(\widehat{\boldsymbol{\theta}}_{\boldsymbol{\lambda}}(z^n)) + \widehat{\mathcal{R}}_{z^n}(\widehat{\boldsymbol{\theta}}_{\boldsymbol{\lambda}}(z^n), \boldsymbol{\lambda}) + O_p\left(\frac{1}{n^2}\right), \tag{12}$$

*where*

$$\widehat{\mathcal{R}}_{z^n}(\boldsymbol{\theta}, \boldsymbol{\lambda}) \triangleq \frac{1}{n(n-1)} \sum_{i \in [n]} \nabla_{\boldsymbol{\theta}}^{\top} \ell(z_i; \boldsymbol{\theta}) \left[\widehat{\mathcal{H}}_{z^{n\backslash i}}(\boldsymbol{\theta}, \boldsymbol{\lambda})\right]^{-1} \nabla_{\boldsymbol{\theta}} \ell(z_i; \boldsymbol{\theta}). \tag{13}$$

Note that in contrast to the ALOOCV (in Theorem 1), the $O_p(1/n^2)$ error term here depends on the second derivative of the loss function with respect to the parameters, consequently leading to worse performance, and underestimation of overfitting.

## 4 Tuning the Regularization Parameters

Thus far, we presented an approximate cross validation vector that closely follows the predictions provided by the cross validation vector, while being computationally inexpensive. In this section, we use the approximate cross validation vector to tune the regularization parameters for the optimal out-of-sample performance. We are interested in solving $\min_{\boldsymbol{\lambda}} \left(\overline{CV}_{\boldsymbol{\lambda}}(z^n) = \frac{1}{n} \sum_{i=1}^{n} \ell\left(z_i; \widehat{\boldsymbol{\theta}}_{\boldsymbol{\lambda}}\left(z^{n\backslash i}\right)\right)\right)$. To this end, we need to calculate the gradient of $\widehat{\boldsymbol{\theta}}_{\boldsymbol{\lambda}}(z^n)$ with respect to $\boldsymbol{\lambda}$, which is given in the following lemma.

**Lemma 4** *We have* $\nabla_{\boldsymbol{\lambda}} \widehat{\boldsymbol{\theta}}_{\boldsymbol{\lambda}}(z^n) = -\frac{1}{n} \left[\widehat{\mathcal{H}}_{z^n}\left(\widehat{\boldsymbol{\theta}}_{\boldsymbol{\lambda}}(z^n), \boldsymbol{\lambda}\right)\right]^{-1} \nabla_{\boldsymbol{\theta}} \mathbf{r}(\widehat{\boldsymbol{\theta}}_{\boldsymbol{\lambda}}(z^n)).$

**Corollary 5** *We have* $\nabla_{\boldsymbol{\lambda}} \widehat{\boldsymbol{\theta}}_{\boldsymbol{\lambda}}(z^{n\backslash i}) = -\frac{1}{n-1} \left[\widehat{\mathcal{H}}_{z^{n\backslash i}}\left(\widehat{\boldsymbol{\theta}}_{\boldsymbol{\lambda}}(z^{n\backslash i}), \boldsymbol{\lambda}\right)\right]^{-1} \nabla_{\boldsymbol{\theta}} \mathbf{r}(\widehat{\boldsymbol{\theta}}_{\boldsymbol{\lambda}}(z^{n\backslash i})).$

In order to apply first order optimization methods for minimizing $\overline{CV}_{\boldsymbol{\lambda}}(z^n)$, we need to compute its gradient with respect to the tuning parameter vector $\boldsymbol{\lambda}$. Applying the simple chain rule implies

$$\nabla_{\boldsymbol{\lambda}} \overline{CV}_{\boldsymbol{\lambda}}(z^n) = \frac{1}{n} \sum_{i=1}^{n} \nabla_{\boldsymbol{\lambda}}^{\top} \widehat{\boldsymbol{\theta}}_{\boldsymbol{\lambda}}(z^{n\backslash i}) \nabla_{\boldsymbol{\theta}} \ell\left(z_i; \widehat{\boldsymbol{\theta}}_{\boldsymbol{\lambda}}\left(z^{n\backslash i}\right)\right) \tag{14}$$

$$= -\frac{1}{n(n-1)} \sum_{i=1}^{n} \nabla_{\boldsymbol{\theta}}^{\top} \mathbf{r}\left(\widehat{\boldsymbol{\theta}}_{\boldsymbol{\lambda}}(z^{n\backslash i})\right) \left[\widehat{\mathcal{H}}_{z^{n\backslash i}}\left(\widehat{\boldsymbol{\theta}}_{\boldsymbol{\lambda}}\left(z^{n\backslash i}\right)\right)\right]^{-1} \nabla_{\boldsymbol{\theta}} \ell\left(z_i; \widehat{\boldsymbol{\theta}}_{\boldsymbol{\lambda}}\left(z^{n\backslash i}\right)\right), \tag{15}$$

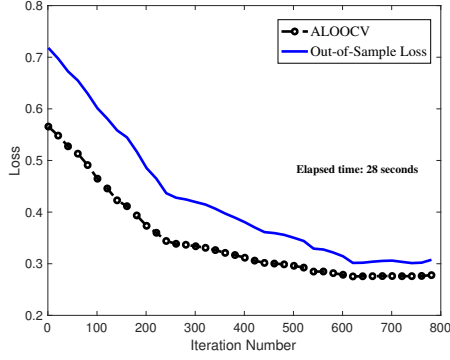

Figure 1: The progression of the loss when Algorithm 1 is applied to ridge regression with diagonal regressors.

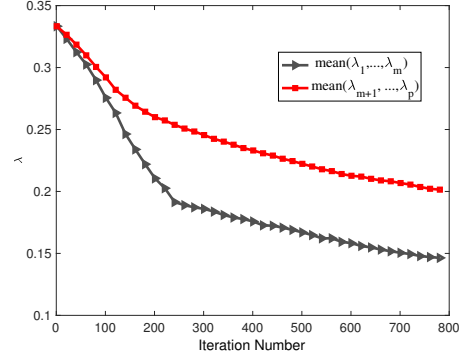

Figure 2: The progression of $\lambda$'s when Algorithm 1 is applied to ridge regression with diagonal regressors.

where (15) follows by substituting $\nabla_{\boldsymbol{\lambda}} \widehat{\boldsymbol{\theta}}_{\boldsymbol{\lambda}}(z^{n\backslash i})$ from Corollary 5. However, (15) is computationally expensive and almost impossible in practice even for medium size datasets. Hence, we use the ALOOCV from (4) (Theorem 1) in (14) to approximate the gradient.

Let

$$\boldsymbol{g}_{\boldsymbol{\lambda}}^{(i)}(z^n) \triangleq -\frac{1}{n-1} \nabla_{\boldsymbol{\theta}}^{\top} \mathbf{r}\left(\widetilde{\boldsymbol{\theta}}_{\boldsymbol{\lambda}}^{(i)}(z^n)\right) \left[\widehat{\mathcal{H}}_{z^{n\backslash i}}\left(\widetilde{\boldsymbol{\theta}}_{\boldsymbol{\lambda}}^{(i)}(z^n)\right)\right]^{-1} \nabla_{\boldsymbol{\theta}}\ell\left(z_i; \widetilde{\boldsymbol{\theta}}_{\boldsymbol{\lambda}}^{(i)}(z^n)\right). \qquad (16)$$

Further, motivated by the suggested ALOOCV, let us define the approximate gradient $\overline{\boldsymbol{g}}_{\boldsymbol{\lambda}}(z^n)$ as $\overline{\boldsymbol{g}}_{\boldsymbol{\lambda}}(z^n) \triangleq \frac{1}{n}\sum_{i\in[n]}\boldsymbol{g}_{\boldsymbol{\lambda}}^{(i)}(z^n)$. Based on our numerical experiments, this approximate gradient closely follows the gradient of the cross validation, i.e., $\nabla_{\boldsymbol{\lambda}}\overline{\mathrm{CV}}_{\boldsymbol{\lambda}}(z^n) \approx \overline{\boldsymbol{g}}_{\boldsymbol{\lambda}}(z^n)$. Note that this approximation is straightforward to compute. Therefore, using this approximation, we can apply the first order optimization algorithm 1 to optimize the tuning parameter $\boldsymbol{\lambda}$. Although Algorithm 1 is

---

**Algorithm 1** Approximate gradient descent algorithm for tuning $\boldsymbol{\lambda}$

---

Initialize the tuning parameter $\boldsymbol{\lambda}^0$, choose a step-size selection rule, and set $t = 0$
**for** $t = 0, 1, 2, \ldots$ **do**
   calculate the approximate gradient $\overline{\boldsymbol{g}}_{\boldsymbol{\lambda}^t}(z^n)$
   set $\boldsymbol{\lambda}^{t+1} = \boldsymbol{\lambda}^t - \alpha^t \overline{\boldsymbol{g}}_{\boldsymbol{\lambda}^t}(z^n)$
**end for**

---

more computationally efficient compared to LOOCV (saving a factor of $n$), it might still be computationally expensive for large values of $n$ as it still scales linearly with $n$. Hence, we also present an online version of the algorithm using the stochastic gradient descent idea; see Algorithm 2.

---

**Algorithm 2** Stochastic (online) approximate gradient descent algorithm for tuning $\boldsymbol{\lambda}$

---

Initialize the tuning parameter $\boldsymbol{\lambda}^0$ and set $t = 0$
**for** $t = 0, 1, 2, \ldots$ **do**
   choose a random index $i_t \in \{1, \ldots, n\}$
   calculate the stochastic gradient $\boldsymbol{g}_{\boldsymbol{\lambda}^t}^{(i_t)}(z^n)$ using (16)
   set $\boldsymbol{\lambda}^{t+1} = \boldsymbol{\lambda}^t - \alpha^t \boldsymbol{g}_{\boldsymbol{\lambda}^t}^{(i_t)}(z^n)$
**end for**

---

## 5 Numerical Experiments

**Ridge regression with diagonal regressors:** We consider the following regularized loss function:

$$w_n(z; \boldsymbol{\theta}, \boldsymbol{\lambda}) = \ell(z; \boldsymbol{\theta}) + \frac{1}{n}\boldsymbol{\lambda}^{\top}\boldsymbol{r}(\boldsymbol{\theta}) = \frac{1}{2}(y - \boldsymbol{\theta}^{\top}x)^2 + \frac{1}{2n}\boldsymbol{\theta}^{\top}\mathrm{diag}(\boldsymbol{\lambda})\boldsymbol{\theta}.$$

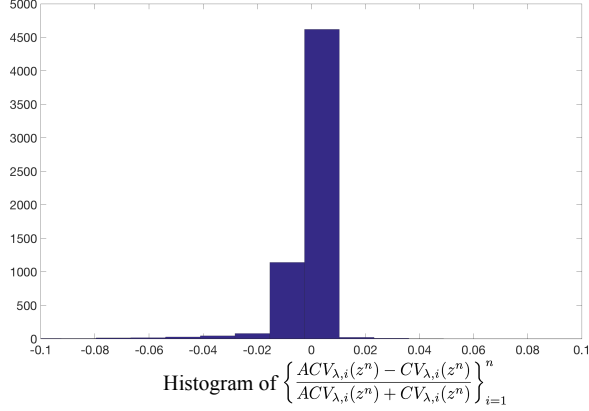

Histogram of $\left\{\frac{ACV_{\lambda,i}(z^n) - CV_{\lambda,i}(z^n)}{ACV_{\lambda,i}(z^n) + CV_{\lambda,i}(z^n)}\right\}_{i=1}^{n}$

Figure 3: The histogram of the normalized difference between LOOCV and ALOOCV for 5 runs of the algorithm on randomly selected samples for each $\lambda$ in Table 1 (MNIST dataset with $n = 200$ and $p = 400$).

| $\lambda$ | $\widehat{\mathcal{L}}_{z^n}$ | $\mathcal{L}$ | CV | ACV | IF |
|---|---|---|---|---|---|
| 3.3333 | 0.0637 (0.0064) | 0.1095 (0.0168) | 0.1077 (0.0151) | 0.1080 (0.0152) | 0.0906 (0.0113) |
| 1.6667 | 0.0468 (0.0051) | 0.1021 (0.0182) | **0.1056** (0.0179) | **0.1059** (0.0179) | 0.0734 (0.0100) |
| 0.8333 | 0.0327 (0.0038) | **0.0996** (0.0201) | 0.1085 (0.0214) | 0.1087 (0.0213) | 0.0559 (0.0079) |
| 0.4167 | 0.0218 (0.0026) | 0.1011 (0.0226) | 0.1158 (0.0256) | 0.1155 (0.0254) | 0.0397 (0.0056) |
| 0.2083 | 0.0139 (0.0017) | 0.1059 (0.0256) | 0.1264 (0.0304) | 0.1258 (0.0300) | 0.0267 (0.0038) |
| 0.1042 | 0.0086 (0.0011) | 0.1131 (0.0291) | 0.1397 (0.0356) | 0.1386 (0.0349) | 0.0171 (0.0024) |
| 0.0521 | **0.0051** (0.0006) | 0.1219 (0.0330) | 0.1549 (0.0411) | 0.1534 (0.0402) | **0.0106** (0.0015) |

Table 1: The results of logistic regression (in-sample loss, out-of-sample loss, LOOCV, and ALOOCV, and Influence Function LOOCV) for different regularization parameters on MNIST dataset with $n = 200$ and $p = 400$. The numbers in parentheses represent the standard error.

| $\frac{n}{p}\lambda$ | $\widehat{\mathcal{L}}_{z^n}$ | $\mathcal{L}$ | ACV |
|---|---|---|---|
| 1e5 | 0.6578 | 0.6591 | 0.6578 (0.0041) |
| 1e4 | 0.5810 | 0.6069 | 0.5841 (0.0079) |
| 1e3 | 0.5318 | 0.5832 | 0.5444 (0.0121) |
| 1e2 | 0.5152 | **0.5675** | **0.5379** (0.0146) |
| 1e1 | 0.4859 | 0.5977 | 0.5560 (0.0183) |
| 1e0 | 0.4456 | 0.6623 | 0.6132 (0.0244) |

Table 2: The results of logistic regression (in-sample loss, out-of-sample loss, CV, ACV) on CIFAR-10 dataset with $n = 9600$ and $p = 3072$.

| $\ell(z_i; \widehat{\boldsymbol{\theta}}_{\boldsymbol{\lambda}}(z^n))$ | CV | ACV | IF |
|---|---|---|---|
| 0.0872 | 8.5526 | 8.6495 | 0.2202 |
| 0.0920 | 2.1399 | 2.1092 | 0.2081 |
| 0.0926 | 10.8783 | 9.4791 | 0.2351 |
| 0.0941 | 3.5210 | 3.3162 | 0.2210 |
| 0.0950 | 5.7753 | 6.1859 | 0.2343 |
| 0.0990 | 5.2626 | 5.0554 | 0.2405 |
| 0.1505 | 12.0483 | 11.5281 | 0.3878 |

Table 3: Comparison of the leave-one-out estimates on the 8 outlier samples with highest in-sample loss in the MNIST dataset.

In other words, we consider one regularization parameter per each model parameter. To validate the proposed optimization algorithm, we consider a scenario with $p = 50$ where $x$ is drawn i.i.d. from $\mathcal{N}(0, \mathbf{I}_p)$. We let $y = \boldsymbol{\theta}^{*\top} x + \epsilon$ where $\theta_1 = \ldots = \theta_{40} = 0$ and $\theta_{41}, \ldots, \theta_{50} \sim \mathcal{N}(0, 1)$ i.i.d, and $\epsilon \sim \mathcal{N}(0, 0.1)$. We draw $n = 150$ samples from this model, and apply Algorithm 1 to optimize for $\boldsymbol{\lambda} = (\lambda_1, \ldots, \lambda_{50})$. The problem is designed in such a way that out of 50 features, the first 40 are irrelevant while the last 10 are important. We initialize the algorithm with $\lambda_1^1 = \ldots = \lambda_{50}^1 = 1/3$ and compute ACV using Theorem 1. Recall that in this case, ACV is exactly equivalent to CV (see Corollary 2). Figure 1 plots ALOOCV, the out-of-sample loss, and the mean value of $\lambda$ calculated over the irrelevant and relevant features respectively. As expected, the $\lambda$ for an irrelevant feature is set to a larger number, on the average, compared to that of a relevant feature. Finally, we remark that the optimization of 50 tuning parameters in 800 iterations took a mere 28 seconds on a PC.

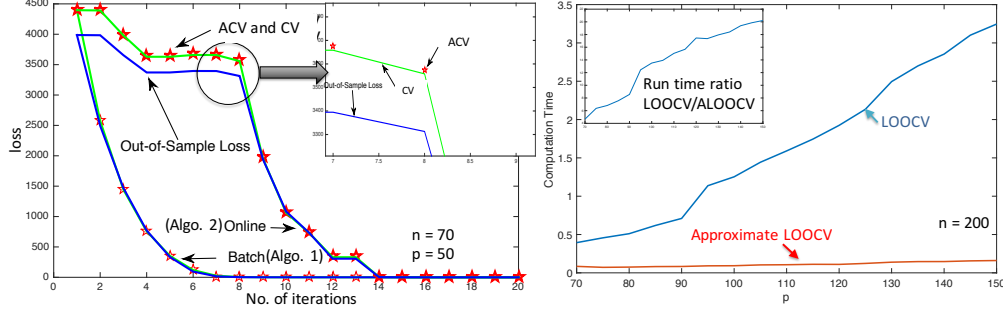

Figure 4: The application of Algorithms 1 and 2 to elastic net regression. The left panel shows the loss vs. number of iterations. The right panel shows the run-time vs. $n$ (the sample size).

**Logistic regression:** The second example that we consider is logistic regression:

$$w_n(z; \boldsymbol{\theta}, \boldsymbol{\lambda}) = \ell(z; \boldsymbol{\theta}) + \frac{1}{n}\boldsymbol{\lambda}^\top \boldsymbol{r}(\boldsymbol{\theta}) = H(y \| \operatorname{sigmoid}(\theta_0 + \boldsymbol{\theta}^\top x)) + \frac{1}{2n}\lambda\|\boldsymbol{\theta}\|_2^2.$$

where $H(\cdot\|\cdot)$ for any $u \in [0,1]$ and $v \in (0,1)$ is given by $H(u\|v) := u \log \frac{1}{v} + (1-u)\log\frac{1}{1-v}$, and denotes the binary cross entropy function, and $\operatorname{sigmoid}(x) := 1/(1 + e^{-x})$ denotes the sigmoid function. In this case, we only consider a single regularization parameter. Since the loss and regularizer are smooth, we resort to Theorem 1 to compute ACV. We applied logistic regression on MNIST and CIFAR-10 image datasets where we used each pixel in the image as a feature according to the aforementioned loss function. In MNIST, we classify the digits 2 and 3 while in CIFAR-10, we classify "bird" and "cat." As can be seen in Tables 1 and 2, ACV closely follows CV on the MNIST dataset. On the other hand, the approximation of LOOCV based on influence functions [14] performs poorly in the regime where the model is significantly overfit and hence it cannot be used for effective model selection. On CIFAR-10, ACV takes $\approx$1s to run per each sample, whereas CV takes $\approx$60s per each sample requiring days to run for each $\lambda$ even for this medium sized problem. The histogram of the normalized difference between CV and ACV vectors is plotted in Figure 3 for 5 runs of the algorithm for each $\lambda$ in Table 1. As can be seen, CV and ACV are almost always within 5% of each other. We have also plotted the loss for the eight outlier samples with the highest in-sample loss in the MNIST dataset in Table 3. As can be seen, ALOOCV closely follows LOOCV even when the leave-one-out loss is two orders of magnitude larger than the in-sample loss for these outliers. On the other hand, the approximation based on the influence functions fails to capture the out-of-sample performance and the outliers in this case.

**Elastic net regression:** Finally, we consider the popular elastic net regression problem [31]:

$$w_n(z; \boldsymbol{\theta}, \boldsymbol{\lambda}) = \ell(z; \boldsymbol{\theta}) + \frac{1}{n}\boldsymbol{\lambda}^\top \boldsymbol{r}(\boldsymbol{\theta}) = \frac{1}{2}(y - \boldsymbol{\theta}^\top x)^2 + \frac{1}{n}\lambda_1\|\boldsymbol{\theta}\|_1 + \frac{1}{2n}\lambda_2\|\boldsymbol{\theta}\|_2^2.$$

In this case, there are only two regularization parameters to be optimized for the quasi-smooth regularized loss. Similar to the previous case, we consider $y = \boldsymbol{\theta}^{*\top} x + \epsilon$ where $\theta_\kappa = \kappa\rho_\kappa\psi_\kappa$ where $\rho_\kappa$ is a Bernoulli(1/2) RV and $\psi_\kappa \sim \mathcal{N}(0,1)$. Hence, the features are weighted non-uniformly in $y$ and half of them are zeroed out on the average. We apply both Algorithms 1 and 2 where we used the approximation in Theorem 1 and the explanation on how to handle $\ell_1$ regularizers to compute ACV. We initialized with $\lambda_1 = \lambda_2 = 0$. As can be seen on the left panel (Figure 4), ACV closely follows CV in this case. Further, we see that both algorithms are capable of significantly reducing the loss after only a few iterations. The right panel compares the run-time of the algorithms vs. the number of samples. This confirms our analysis that the run-time of CV scales quadratically with $O(n^2)$ as opposed to $O(n)$ in ACV. This impact is more signified in the inner panel where the run-time ratio is plotted.

## Acknowledgement

This work was supported in part by DARPA under Grant No. W911NF-16-1-0561. The authors are thankful to Jason D. Lee (USC) who brought to their attention the recent work [14] on influence functions for approximating leave-one-out cross validation.

## Footnotes

[3] $(\cdot)^\circ$ denotes the interior operator.

[4] $X_n = o_p(a_n)$ implies that $X_n/a_n$ approaches 0 in probability with respect to the density function $p(\cdot)$.

[5] $X_n = O_p(a_n)$ implies that $X_n / a_n$ is stochastically bounded with respect to the density function $p(\cdot)$.

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
