[Supplementary Material · nips2017_ACV_FINAL_appendix.pdf]

# Appendix

**Proof of Theorem 1** Let us proceed with some algebraic manipulations. As the regularized loss function is assumed to be of class $C^3$ in a neighborhood of the solution, invoking Taylor's theorem notice that

$$\nabla_{\boldsymbol{\theta}} w_{n-1}(z_j; \widehat{\boldsymbol{\theta}}_{\boldsymbol{\lambda}}(z^n), \boldsymbol{\lambda}) = \nabla_{\boldsymbol{\theta}} w_{n-1}(z_j; \widehat{\boldsymbol{\theta}}_{\boldsymbol{\lambda}}(z^{n \setminus i}), \boldsymbol{\lambda}) \tag{17a}$$

$$+ \nabla_{\boldsymbol{\theta}}^2 w_{n-1}(z_j; \widehat{\boldsymbol{\theta}}_{\boldsymbol{\lambda}}(z^{n \setminus i}), \boldsymbol{\lambda})(\widehat{\boldsymbol{\theta}}_{\boldsymbol{\lambda}}(z^n) - \widehat{\boldsymbol{\theta}}_{\boldsymbol{\lambda}}(z^{n \setminus i})) \tag{17b}$$

$$+ \frac{1}{2} \sum_{\kappa \in [k]} (\widehat{\boldsymbol{\theta}}_{\boldsymbol{\lambda}}(z^n) - \widehat{\boldsymbol{\theta}}_{\boldsymbol{\lambda}}(z^{n \setminus i}))^\top \left( \frac{\partial}{\partial \theta_\kappa} \nabla_{\boldsymbol{\theta}}^2 w_{n-1}(z_j; \boldsymbol{\zeta}_{\boldsymbol{\lambda}, \kappa}^{i,j,1}(z^n), \boldsymbol{\lambda}) \right) (\widehat{\boldsymbol{\theta}}_{\boldsymbol{\lambda}}(z^n) - \widehat{\boldsymbol{\theta}}_{\boldsymbol{\lambda}}(z^{n \setminus i})) \widehat{e}_\kappa \tag{17c}$$

where $\boldsymbol{\zeta}_{\boldsymbol{\lambda}, \kappa}^{i,j,1}(z^n) = \alpha_\kappa^{i,j,1} \widehat{\boldsymbol{\theta}}_{\boldsymbol{\lambda}}(z^n) + (1 - \alpha_\kappa^{i,j,1}) \widehat{\boldsymbol{\theta}}_{\boldsymbol{\lambda}}(z^{n \setminus i})$ for some $0 \leq \alpha_\kappa^{i,j,1} \leq 1$. Hence, we can sum both sides up over $j \in [n] \setminus i$ to get

$$\sum_{j \in [n] \setminus i} \nabla_{\boldsymbol{\theta}} w_{n-1}(z_j; \widehat{\boldsymbol{\theta}}_{\boldsymbol{\lambda}}(z^n), \boldsymbol{\lambda}) = \sum_{j \in [n] \setminus i} \nabla_{\boldsymbol{\theta}} w_{n-1}(z_j; \widehat{\boldsymbol{\theta}}_{\boldsymbol{\lambda}}(z^{n \setminus i}), \boldsymbol{\lambda}) \tag{18a}$$

$$+ \sum_{j \in [n] \setminus i} \nabla_{\boldsymbol{\theta}}^2 w_{n-1}(z_j; \widehat{\boldsymbol{\theta}}_{\boldsymbol{\lambda}}(z^{n \setminus i}), \boldsymbol{\lambda})(\widehat{\boldsymbol{\theta}}_{\boldsymbol{\lambda}}(z^n) - \widehat{\boldsymbol{\theta}}_{\boldsymbol{\lambda}}(z^{n \setminus i})) \tag{18b}$$

$$+ \varepsilon_{\boldsymbol{\lambda}, n}^{(i), 1}, \tag{18c}$$

where $\varepsilon_{\boldsymbol{\lambda}, n}^{(i), 1}$ is defined in (7). Notice that by definition of $\widehat{\boldsymbol{\theta}}_{\boldsymbol{\lambda}}(z^n)$ and $\widehat{\boldsymbol{\theta}}_{\boldsymbol{\lambda}}(z^{n \setminus i})$, the left hand side term in (18a) is equal to $-\nabla_{\boldsymbol{\theta}} \ell(z_i; \widehat{\boldsymbol{\theta}}_{\boldsymbol{\lambda}}(z^n))$ and the right hand side term is zero. Then,

$$-\nabla_{\boldsymbol{\theta}} \ell(z_i; \widehat{\boldsymbol{\theta}}_{\boldsymbol{\lambda}}(z^n)) = \sum_{j \in [n] \setminus i} \nabla_{\boldsymbol{\theta}}^2 w_{n-1}(z_j; \widehat{\boldsymbol{\theta}}_{\boldsymbol{\lambda}}(z^{n \setminus i}), \boldsymbol{\lambda})(\widehat{\boldsymbol{\theta}}_{\boldsymbol{\lambda}}(z^n) - \widehat{\boldsymbol{\theta}}_{\boldsymbol{\lambda}}(z^{n \setminus i})) + \varepsilon_{\boldsymbol{\lambda}, n}^{(i), 1}. \tag{19}$$

Applying Taylor's theorem on $\nabla_{\boldsymbol{\theta}}^2 w_{n-1}(z_j; \widehat{\boldsymbol{\theta}}_{\boldsymbol{\lambda}}(z^{n \setminus i}), \boldsymbol{\lambda})$ we get:

$$\nabla_{\boldsymbol{\theta}}^2 w_{n-1}(z_j; \widehat{\boldsymbol{\theta}}_{\boldsymbol{\lambda}}(z^{n \setminus i}), \boldsymbol{\lambda}) = \nabla_{\boldsymbol{\theta}}^2 w_{n-1}(z_j; \widehat{\boldsymbol{\theta}}_{\boldsymbol{\lambda}}(z^n), \boldsymbol{\lambda})$$

$$+ \sum_{\kappa, \nu \in [k]} (\widehat{\boldsymbol{\theta}}_{\boldsymbol{\lambda}}(z^{n \setminus i}) - \widehat{\boldsymbol{\theta}}_{\boldsymbol{\lambda}}(z^n))^\top \left( \frac{\partial^2}{\partial \theta_\kappa \partial \theta_\nu} \nabla_{\boldsymbol{\theta}} w_{n-1}(z_j; \boldsymbol{\zeta}_{\boldsymbol{\lambda}, \kappa, \nu}^{i,j,2}(z^n), \boldsymbol{\lambda}) \right) \widehat{e}_\kappa \widehat{e}_\nu^\top. \tag{20}$$

By substituting (20) in (19), using some algebraic manipulations, and noting the definition of $\varepsilon_{\boldsymbol{\lambda}, n}^{i,j,2}$ in (8), we can get

$$-\nabla_{\boldsymbol{\theta}} \ell(z_i; \widehat{\boldsymbol{\theta}}_{\boldsymbol{\lambda}}(z^n)) = \sum_{j \in [n] \setminus i} \nabla_{\boldsymbol{\theta}}^2 w_{n-1}(z_j; \widehat{\boldsymbol{\theta}}_{\boldsymbol{\lambda}}(z^n), \boldsymbol{\lambda})(\widehat{\boldsymbol{\theta}}_{\boldsymbol{\lambda}}(z^n) - \widehat{\boldsymbol{\theta}}_{\boldsymbol{\lambda}}(z^{n \setminus i})) + \varepsilon_{\boldsymbol{\lambda}, n}^{(i)}, \tag{21}$$

where $\varepsilon_{\boldsymbol{\lambda}, n}^{(i)}$ is defined in (6). Consequently,

$$\widehat{\boldsymbol{\theta}}_{\boldsymbol{\lambda}}(z^{n \setminus i}) - \widehat{\boldsymbol{\theta}}_{\boldsymbol{\lambda}}(z^n) = \left( \sum_{j \in [n] \setminus i} \nabla_{\boldsymbol{\theta}}^2 w_{n-1}(z_j; \widehat{\boldsymbol{\theta}}_{\boldsymbol{\lambda}}(z^n), \boldsymbol{\lambda}) \right)^{-1} \nabla_{\boldsymbol{\theta}} \ell(z_i; \widehat{\boldsymbol{\theta}}_{\boldsymbol{\lambda}}(z^n)) + \boldsymbol{\xi}_{\boldsymbol{\lambda}, n}^{(i)} \tag{22}$$

$$= \frac{1}{n-1} \left( \widehat{\mathcal{H}}_{z^{n \setminus i}}(\widehat{\boldsymbol{\theta}}_{\boldsymbol{\lambda}}(z^n), \boldsymbol{\lambda}) \right)^{-1} \nabla_{\boldsymbol{\theta}} \ell(z_i; \widehat{\boldsymbol{\theta}}_{\boldsymbol{\lambda}}(z^n)) + \boldsymbol{\xi}_{\boldsymbol{\lambda}, n}^{(i)}, \tag{23}$$

where

$$\boldsymbol{\xi}_{\boldsymbol{\lambda}, n}^{(i)} \triangleq \frac{1}{n-1} \left( \widehat{\mathcal{H}}_{z^{n \setminus i}}(\widehat{\boldsymbol{\theta}}_{\boldsymbol{\lambda}}(z^n), \boldsymbol{\lambda}) \right)^{-1} \varepsilon_{\boldsymbol{\lambda}, n}^{(i)}. \tag{24}$$

Note that Assumption 1 implies that as $n$ grows, the above inverse with high probability exists and converges to $\mathcal{H}(\boldsymbol{\theta}^*)^{-1}$ in probability. Further, the inverse is bounded in probability.

Finally, it is deduced from Assumption 1 that $\|\widehat{\boldsymbol{\theta}}_{\boldsymbol{\lambda}}(z^n) - \widehat{\boldsymbol{\theta}}_{\boldsymbol{\lambda}}(z^{n\backslash i}))\|_\infty = o_p(1)$. On the other hand, by noticing the definition of $\varepsilon_{\boldsymbol{\lambda},n}^{(i)}$, we can see that

$$\|\varepsilon_{\boldsymbol{\lambda},n}^{(i)}\|_\infty = O_p\left(n\|\widehat{\boldsymbol{\theta}}_{\boldsymbol{\lambda}}(z^n) - \widehat{\boldsymbol{\theta}}_{\boldsymbol{\lambda}}(z^{n\backslash i}))\|_\infty^2\right) = o_p\left(n\|\widehat{\boldsymbol{\theta}}_{\boldsymbol{\lambda}}(z^n) - \widehat{\boldsymbol{\theta}}_{\boldsymbol{\lambda}}(z^{n\backslash i}))\|_\infty\right). \tag{25}$$

Hence, considering (24),

$$\|\boldsymbol{\xi}_{\boldsymbol{\lambda},n}^{(i)}\|_\infty = o_p\left(\|\widehat{\boldsymbol{\theta}}_{\boldsymbol{\lambda}}(z^n) - \widehat{\boldsymbol{\theta}}_{\boldsymbol{\lambda}}(z^{n\backslash i}))\|_\infty\right). \tag{26}$$

Now, considering (23), we deduce that

$$\|\widehat{\boldsymbol{\theta}}_{\boldsymbol{\lambda}}(z^n) - \widehat{\boldsymbol{\theta}}_{\boldsymbol{\lambda}}(z^{n\backslash i}))\|_\infty = O_p\left(\frac{1}{n}\right). \tag{27}$$

Hence, $\|\varepsilon_{\boldsymbol{\lambda},n}^{(i)}\|_\infty = O_p(1/n)$ in (6). Thus, the error term is

$$\widehat{\boldsymbol{\theta}}_{\boldsymbol{\lambda}}(z^{n\backslash i}) - \widetilde{\boldsymbol{\theta}}_{\boldsymbol{\lambda}}^{(i)}(z^n) = \boldsymbol{\xi}_{\boldsymbol{\lambda},n}^{(i)}, \tag{28}$$

and

$$\|\widehat{\boldsymbol{\theta}}_{\boldsymbol{\lambda}}(z^{n\backslash i}) - \widetilde{\boldsymbol{\theta}}_{\boldsymbol{\lambda}}^{(i)}(z^n)\|_\infty = O_p\left(\frac{1}{n^2}\right), \tag{29}$$

completing the proof. ∎

**Proof of Lemma 3** Notice that

$$
\begin{aligned}
\ell(z_i; \widetilde{\boldsymbol{\theta}}_{\boldsymbol{\lambda}}^{(i)}(z^n))) =& \ell(z_i; \widehat{\boldsymbol{\theta}}_{\boldsymbol{\lambda}}(z^n)) \\
&+ \nabla_{\boldsymbol{\theta}}^\top \ell(z_i; \widehat{\boldsymbol{\theta}}(z^n))(\widetilde{\boldsymbol{\theta}}_{\boldsymbol{\lambda}}^{(i)}(z^n)) - \widehat{\boldsymbol{\theta}}_{\boldsymbol{\lambda}}(z^n)) \\
&+ O(\|\widetilde{\boldsymbol{\theta}}_{\boldsymbol{\lambda}}^{(i)}(z^n)) - \widehat{\boldsymbol{\theta}}_{\boldsymbol{\lambda}}(z^n)\|_\infty^2) \\
=& \ell(z_i; \widehat{\boldsymbol{\theta}}_{\boldsymbol{\lambda}}(z^n)) \\
&+ \frac{1}{n-1}\nabla_{\boldsymbol{\theta}}^\top \ell(z_i; \widehat{\boldsymbol{\theta}}_{\boldsymbol{\lambda}}(z^n)) \left[\widehat{\mathcal{H}}_{z^{n\backslash i}}(\widehat{\boldsymbol{\theta}}_{\boldsymbol{\lambda}}(z^n), \boldsymbol{\lambda})\right]^{-1} \nabla_{\boldsymbol{\theta}}\ell(z_i; \widehat{\boldsymbol{\theta}}_{\boldsymbol{\lambda}}(z^n)) \\
&+ O_p\left(\frac{1}{n^2}\right),
\end{aligned}
$$

$$\tag{30}$$
$$\tag{31}$$

where (30) follows from Assumption 1, and (31) follows from the definition of $\widetilde{\boldsymbol{\theta}}_{\boldsymbol{\lambda}}^{(i)}(z^n)$, and the fact that

$$\|\widetilde{\boldsymbol{\theta}}_{\boldsymbol{\lambda}}^{(i)}(z^n) - \widehat{\boldsymbol{\theta}}_{\boldsymbol{\lambda}}(z^n)\|_\infty \leq \|\widehat{\boldsymbol{\theta}}_{\boldsymbol{\lambda}}(z^{n\backslash i}) - \widetilde{\boldsymbol{\theta}}_{\boldsymbol{\lambda}}^{(i)}(z^n)\|_\infty + \|\widehat{\boldsymbol{\theta}}_{\boldsymbol{\lambda}}(z^n) - \widehat{\boldsymbol{\theta}}_{\boldsymbol{\lambda}}(z^{n\backslash i}))\|_\infty \tag{32}$$

$$= O_p\left(\frac{1}{n^2}\right) + O_p\left(\frac{1}{n}\right) = O_p\left(\frac{1}{n}\right), \tag{33}$$

which implies $\|\widehat{\boldsymbol{\theta}}_{\boldsymbol{\lambda}}(z^n) - \widetilde{\boldsymbol{\theta}}_{\boldsymbol{\lambda}}^{(i)}(z^n)\|_\infty^2 = O_p(1/n^2)$. The proof is completed by noticing the definition of $\overline{\mathrm{ACV}}_{\boldsymbol{\lambda}}(z^n)$ in (11). ∎

**Proof of Lemma 4** By definition,

$$\nabla_{\boldsymbol{\theta}}\widehat{\mathcal{L}}_{z^n}(\widehat{\boldsymbol{\theta}}_{\boldsymbol{\lambda}}(z^n)) + \frac{1}{n}\nabla_{\boldsymbol{\theta}}\mathbf{r}(\widehat{\boldsymbol{\theta}}_{\boldsymbol{\lambda}}(z^n))\boldsymbol{\lambda} = \nabla_{\boldsymbol{\theta}}\widehat{\mathcal{W}}_{z^n}(\widehat{\boldsymbol{\theta}}_{\boldsymbol{\lambda}}(z^n)) = 0. \tag{34}$$

Using the implicit function theorem, we can further differentiate the left-hand-side with respect to $\boldsymbol{\lambda}$ to get:

$$\nabla_{\boldsymbol{\theta}}^2\widehat{\mathcal{L}}_{z^n}(\widehat{\boldsymbol{\theta}}_{\boldsymbol{\lambda}}(z^n))\nabla_{\boldsymbol{\lambda}}\widehat{\boldsymbol{\theta}}_{\boldsymbol{\lambda}}(z^n) + \frac{1}{n}\nabla_{\boldsymbol{\theta}}\mathbf{r}(\widehat{\boldsymbol{\theta}}_{\boldsymbol{\lambda}}(z^n)) + \frac{1}{n}\sum_{m\in[M]}\lambda_m\nabla_{\boldsymbol{\theta}}^2 r_m(\widehat{\boldsymbol{\theta}}_{\boldsymbol{\lambda}}(z^n))\nabla_{\boldsymbol{\lambda}}\widehat{\boldsymbol{\theta}}_{\boldsymbol{\lambda}}(z^n) = 0.$$

$$\tag{35}$$

Thus,

$$\widehat{\mathcal{H}}_{z^n}\left(\widehat{\boldsymbol{\theta}}_{\boldsymbol{\lambda}}(z^n), \boldsymbol{\lambda}\right)\nabla_{\boldsymbol{\lambda}}\widehat{\boldsymbol{\theta}}_{\boldsymbol{\lambda}}(z^n) = -\frac{1}{n}\nabla_{\boldsymbol{\theta}}\mathbf{r}(\widehat{\boldsymbol{\theta}}_{\boldsymbol{\lambda}}(z^n)), \tag{36}$$

which completes the proof. ∎

**Proof of Corollary 5** This directly follows from Lemma 4. ∎