[Reviews · NeurIPS 2017]

Reviewer 1



As is well known, the leave-one-out CV (LOOCV) error can be efficiently computed for ridge regression, thanks to efficient formulas for adjusting the loss function value when one sample point is removed. The authors propose an approximate formula for LOOCV for other models by generalizing this idea. I haven't seen this proposed before. Looks like the idea has plenty of potential. The paper is really well written and polished, and appears technically solid. I could't find any typos or other minor details to fix, except: - references: please use capital initials in "Bayesian" and for all words in book and journal titles

Reviewer 2



This paper proposes an efficiently computable approximation of leave-one-out cross validation for parametric learning problems, as well as an algorithm for jointly learning the regularization parameters and model parameters. These techniques seem novel and widely applicable. The paper starts out clearly written, though maybe some less space could have been spent on laying the groundwork, leaving more room for the later sections where the notation is quite dense. Can you say anything about the comparsion between ALOOCV and LOOCV evaluated on only a subset of the data points (as you mention in l137-140), both in terms of computation cost and approximation accuracy? Other comments: l75: are you referring to PRESS? Please name it then. l90: "no assumptions on the distribution" -- does that mean, no prior distribution? Definition 7: the displayed equation seems to belong to the "such that" in the preceding sentence; please pull it into the same sentence. Also, I find it odd that an analytic function doesn't satisfy this definition (due to the "there exists one and only one"). What about a two-dimensional function that has non-differentiabilities in its uppper-right quadrant, so that along some cross sections, it is analytic? l186-187: (relating to the remarks about Def7) This sounds a bit odd; it might be better to say something like "We remark that the theory could be extended to ...". l250: Are you saying that you are not jointly learning the regularization parameter in this second example? If the material in section 4 doesn't apply here, I missed why; please clarify in that section. Typographic: l200: few -> a few References: ensure capitalization of e.g. Bayes by using {}